# Study on the Influence of Erbium and Preheating Process on Mechanical Properties of As-Cast 7055 Aluminum Alloy

**DOI:** 10.3390/ma16155296

**Published:** 2023-07-27

**Authors:** Jingwei Li, Faguo Li

**Affiliations:** School of Materials Science and Engineering, Xiangtan University, Xiangtan 411105, China; 202005721027@smail.xtu.edu.cn

**Keywords:** 7055 aluminum alloy, erbium, mechanical properties, microstructure

## Abstract

Although 7055 aluminum alloy is a deformed aluminum alloy and shows excellent mechanical properties after recrystallization and large deformation, through this method, its application range is enriched if rare earth is added, and the rare earth phase dispersion is promoted by heat treatment. This article used optical microscopy, scanning electron microscopy energy dispersive spectroscopy (SEM-EDS), X-ray diffraction (XRD), micro Vickers hardness, and room temperature stretching methods to study the as-cast 7055-*x*Er (*x* = 0 wt.%, 0.2 wt.%, 0.4 wt.%, 0.6 wt.%, 0.8 wt.%) aluminum alloy after being subjected to 460 °C × 3 h homogenization and 410 °C × 1 h solid solution + 150 °C × 12 h aging treatment for the changes in its microstructure and properties. The results indicated that: when 0.2 wt.%Er was added to 7055 aluminum alloy after a solution at 410 °C × 1 h and aging at 150 °C × 12 h, the dendrite structure was significantly reduced, the grain thinning was obvious, and the distribution was uniform; the Al_8_Cu_4_Er phase appeared in the lamellar eutectic *η*-Mg(Zn,Al,Cu)_2_ structure at grain boundaries, and the hardness reached 168.8 HV. The yield strength, tensile strength, and elongation were 542.12 MPa, 577.67 MPa, and 8.36%, respectively.

## 1. Introduction

The 7055 aluminum alloy not only has high strength but also has the advantages of low density and excellent processability, which is widely used in aerospace. It is often used in the manufacture of aircraft fuselage, wing beams, and other structural products [1,2]. In recent years, with the continuous improvement of energy conservation and emission reduction requirements, low density and high strength have become the mainstream pursuit of aluminum alloy [3]. The conventional 7xxx alloy is a typical deformed aluminum alloy with excellent mechanical properties [4]. However, the deformation process is long and complicated, which increases the production cost and limits its application field. Cast aluminum alloys have the advantages of a short process flow, near-net forming, and low cost [5]. Their biggest consumer is the automobile industry, which is used in various parts, such as the engine block and gearbox of internal combustion engines. It can also be used in the manufacture of an automobile chassis and body [6]. In order to improve the performance of cast aluminum alloys to meet the higher requirements of mechanical devices, heat treatment [7] and microalloying [8,9,10] can be used.

Because of its unique electronic shell structure, rare earth elements can play a unique role in smelting and alloying in aluminum alloy production and can also play an important role in the development of an ultra-strong aluminum alloy [7,8,9]. The addition of trace Sc in Al-Cu alloy can promote *θ*′ precipitation in grain and inhibit *θ*′ precipitation at the grain boundary while also inhibiting the transformation of *θ*′ into *θ* phase. There is an obvious strengthening effect [11,12,13,14]. The price of Sc is too high, and even a small amount of the addition can greatly increase the commercialization cost. The lower-priced Er can also strengthen the 7055 aluminum alloy [10], becoming a potential rare earth-modified element of 7055 aluminum alloy. The addition of Er produces Al_3_Er particles, which can inhibit the migration of subgrain boundaries during the solution aging of aluminum alloy, thus retaining more small angle grain boundaries (LAGBs). The greater the proportion of LAGBs in the alloy, the better the corrosion resistance of the alloy [8,15]. The composite addition of Er, Cr, and Zr can also enhance the recrystallization performance and stabilize the deformation recovery structure with many fine subgrain boundaries [10]. Moreover, the synergistic effect of Si, Zr, and Er elements after Er was added to 7055 aluminum alloy inhibited the formation of hot cracks during pouring and prevented the appearance of impurities and pores [16].

The heat treatment process has an important effect on improving the properties of 7055 aluminum alloy [17]. As a kind of age-hardening aluminum alloy, the excellent properties of the 7055 alloy are mainly attributed to its uniform eutectic distribution Guinier–Preston zones (GP zones), the semi-coherent *η*′ phase, and incoherent *η* phase. The precipitation sequence of these phases is as follows [18]: the supersaturated solid solution zone (SSSS)-G.P. zones-*η*′ phase (metastable MgZn_2_)-*η* phase (equilibrium MgZn_2_). Adjusting the size and quantity of the GP zone and *η*′ phase formed in the early aging period is the key to improving the performance. At present, it is believed that the main strengthening effect of the Al-Zn-Mg-Cu alloy is the joint action of the GP region and *η*′, and some scholars believe that the main strengthening phase is *η*′ phase [19,20]. Precipitation hardening occurs in certain steps, such as solid solution treatment at the appropriate temperature and time and rapid cooling by quenching and aging treatment for a certain time to form a second phase of precipitation inside the matrix [21]. Solution treatment is an important part of heat treatment, which is carried out at a relatively high temperature to dissolve the secondary phase formed during the solidification process so that the alloying elements in the solid solution concentration are high and uniform. In general, for most of the as-cast 7055 aluminum alloy, the solution temperature is selected at 450–470 °C [22,23,24], while the Zn phase has a lower melting point (about 419.5 °C). A higher solution temperature can lead to overheating [25] and form a thin and brittle oxide film at the banded interface [26]. Aging is the process of decomposition for the supersaturated solid solution and precipitation of the strengthened phase. According to research [27], the as-cast 7xxx series aluminum alloy can reach a peak hardness of 189 HV after aging at 120 °C for 156 h, and the yield strength increases by 240 MPa compared with that of the as-cast aluminum alloy. After aging at 120 °C × 1 h + 150 °C × 4 h, the time consumption is reduced, the hardness peak is reached at 184.7 HV, and the yield strength is increased by 235 MPa compared with the as-cast condition. Increasing the aging temperature to 150 °C accelerates the growth of the GP region, shortens the time to reach the peak hardness, and obtains considerable mechanical properties. A higher aging temperature or longer aging time can increase the volume fraction of the GP region of the alloy, improve the strength of the alloy, and obviously prevent the ductility from deteriorating [28]. The single-stage aging temperature is generally 120–180 °C [29,30,31].

In order to refine the alloy structure and obtain the dispersed second phase to enhance the properties of the 7055 aluminum alloy, it should be considered that excessive temperatures can lead to the coarsening of the second phase size. In order to avoid this phenomenon and effectively improve the microstructure and mechanical properties of the material with a rare earth phase, the solid solution treatment is set at 410 °C × 1 h. Combined with the above analysis, the aging treatment was set at 150 °C × 12 h. At present, most articles have only pointed out that the addition of a small amount of Er can refine aluminum alloy; however, the heat treatment process of as-cast 7055 aluminum alloy containing Er has not been studied in detail. In this study, the process path of adding the trace erbium element and heat treatment to as-cast 7055 aluminum alloy is proposed to further refine the grain structure of 7055 aluminum alloy and improve the strength and toughness of 7055 aluminum alloy. The microstructure and mechanical properties of the as-cast 7055 aluminum alloy, as-cast heat-treated 7055 aluminum alloy, as-cast 7055 aluminum alloy containing Er, and as-cast heat-treated 7055 aluminum alloy containing Er were comparatively studied, and the influence mechanism of Er and heat treatment process on as-cast 7055 aluminum alloy is discussed.

## 2. Materials and Methods

### 2.1. Preparation of Experimental Materials

7055 aluminum alloy and Er (99.99 wt.%) were used as raw materials, and the 7055 aluminum alloy composition was Al-7.9Zn-2.5Cu-2.1Mg-0.1Zr-0.1Fe (wt.%). An electronic balance (Shanghai Huachao Industrial Co., Ltd., Shanghai, China) was used to weigh good raw materials according to the ratio of ingredients, which was made accurate to four decimal places. The 7055 aluminum alloy was loaded into the alumina crucible at 780 °C and completely melted in the intelligent electric heating equipment (Xiangtan Samsung Instrument Co., Ltd., Xiangtan, China). Then, the rare earth element was added Er wrapped in aluminum foil, and melted for 15 min, stirring with ceramic bar when added so that Er was more evenly integrated into the 7055 aluminum alloy. Then, the temperature was reduced to 740 °C for 15 min so that the gas in the solution could be completely eliminated to prevent pores and slag inclusion in the casting. Finally, the metal liquid was poured into the graphite mold along the edge of the graphite mold. It cooled naturally to room temperature and was removed from the mold.

The sample was homogenized in a tube furnace (Xiangtan Samsung Instrument Co., Ltd., Xiangtan, China) at 460 °C × 3 h. The as-cast 7055-*x*Er (*x* = 0 wt.%, 0.2 wt.%, 0.4 wt.%, 0.6 wt.%, 0.8 wt.%) was obtained. The cast alloy was heat treated; the tube furnace was set at the solution temperature of 410 °C; the sample was immediately put into water to cool after holding for 1 h. Then, it was put into the tube furnace with the temperature set at 150 °C for 12 h aging, and the heat treated 7055-*x*Er aluminum alloy was obtained.

### 2.2. Data Acquisition and Analysis Methods

OM (ZEISS, ZEISS, Jena, Germany), SEM-EDS (ZEISS, EVO MA10, ZEISS, Jena, Germany), XRD (Ultima IV, Rigaku Co., Tokyo, Japan) grain size measurement and precipitated phase analysis were carried out, respectively. The Vickers hardness tester (SHYCHVT-30, Laizhou Huayin Hardness Meter Factory, Laizhou, Shandong, China) measured the hardness of the sample.

When measuring the size of the crystal phase, after grinding and polishing to a mirror surface, the corrosion was carried out with Keller’s etch (95 mL water, 2.5 mLHNO_3_, 1.5 mLHCL, 1.0 mLHF). Metallographic photographs were taken at five different locations of each sample, and the grain size was measured using the intercept method. Image-Pro Plus v6.0 software was used to process metallographic photos. First, a straight-line segment was drawn in the metallographic photo, and the length of the line segment was recorded. Then, the number of grains passing through the line segment was counted before the length of the line segment was divided by the number of grains to obtain the average diameter of each grain. The average grain diameter in each metallographic photograph was measured 3 times. Finally, the average of the results obtained from all 5 metallographic photographs was taken as the grain diameter value of the sample.

Hardness measurement method: After pre-grinding and polishing, the Vickers hardness was measured, and the average value of the three points was taken. The test load was 5000 g, and the residence time of the indenter was 15 s. The tensile test was carried out at room temperature using an electronic multifunctional testing machine (WDW-100C, Jinan Fangyuan Instrument Co., Ltd., Jinan, China). Five parallel samples were taken for each sample. The specimen thickness was 1.5 mm, the tensile rate was 0.1 mm/min, and the tensile specimen was plate-like. In addition, all specimens tested for mechanical properties were screened by visual and optical microscopy. Then, density screening was conducted: the weight of the tensile specimen was first measured with an electronic balance. Then, the specimen was immersed in a measuring cup with water, and the volume of the specimen was calculated. Then, their density was calculated. Samples that differed from the theoretical density value by ±0.5% were retained.

The phase composition of the sample was analyzed by XRD, the scanning rate was set to 10°/min, and the X-ray diffraction range was 10–90°. When SEM-EDS analyzed the chemical composition, the sample used for the observation and analysis of the second phase was ground and polished, similar to the preparation of the metallographic sample before observation and without metallographic corrosion.

## 3. Results

### 3.1. Microstructure and Mechanical Properties

Figure 1 shows the metallographic microstructure (OM) of the as-cast 7055 aluminum alloy and as-cast heat-treated 7055-*x*Er aluminum alloy. The dendrite coarsening state of the as-cast 7055 aluminum alloy was obviously improved after heat treatment, and the grain was refined. After the heat treatment of the 7055-0.2 wt.%Er aluminum alloy, the grain was obviously refined and appeared as a cell crystal. However, the content of the Er element also increased to 0.4 wt.%, 0.6 wt.%, and 0.8 wt.% after heat treatment and coarse dendrites appeared again. When the content of Er exceeded a certain value, since aluminum is a face-centered cubic structure, Er formed a close-packed hexagonal structure. The atomic radius of Er (0.176 nm) is larger than that of Al (0.143 nm). If Er enters the crystal lattice of Al, it causes large lattice distortion and increases the system energy. In order to reduce the grain boundary energy, Er can be enriched toward the grain boundary. During solidification, Er gathers at the front of the solid–liquid interface, increasing the constitutional supercoiling [32]. This then promotes nucleation and a reduction in grain size. Therefore, the proper addition of Er can promote grain refinement.

### 3.2. Phase Component Analysis

The phase composition of as-cast 7055 aluminum alloy, as-cast 7055-0.2 wt.%Er, and as-cast heat-treated 7055-0.2 wt.%Er was analyzed by XRD is shown in Figure 2. The XRD pattern of the as-cast 7055 aluminum alloy is similar to that of the as-cast 7055 aluminum alloy; therefore, it is not marked. The main precipitated phases of the 7055-aluminum alloy before and after heat treatment were *α*-Al, *η*-MgZn_2_. The diffraction peak of Al_8_Cu_4_Er (PDF 33-0006) appeared near 29° when Er was added after heat treatment (see Figure 1b).

Figure 3 shows the SEM backscattering diagram of the 7055 aluminum alloy with different components after cast and heat treatment, and the EDS measurement results of the corresponding intermetallic compounds on the diagram are shown in Table 1. Intermetallic compounds appear brighter in the image because their atomic number was higher than that of the Al matrix. As-cast Al-Mg-Zn-Cu alloys are generally composed of an α(Al) matrix and non-equilibrium eutectic structures, including *α*-Al, *η*-Mg (Zn,Al,Cu)_2_, S-CuMgAl_2_, Al_7_Cu_2_Fe and Mg_2_Si [33,34,35]. EDS analysis showed that high-density lamellar eutectic *η*-Mg(Zn,Al,Cu)_2_ existed in all four samples (Figure 3 point A, B, D, F). In the 7055-0.2%Er aluminum alloy, a new phase Al_8_Cu_4_Er (Figure 3 point C, E) appeared in both the as-cast and solid-solution aging states in the form of polygonal blocks. In addition, Al_3_Er may also exist; however, due to the small content, it was not detected by XRD.

Figure 4 shows the mapping scanning results of the 7055 aluminum alloy with different components after casting and heat treatment. Compared with the original as-cast 7055 aluminum alloy, the Mg and Zn segregation of the 7055 aluminum alloy with Er element was relatively weakened (Figure 4a–d). After heat treatment, the Er element in the 7055-0.2 wt.%Er aluminum alloy was more evenly dispersed than that in the as-cast 7055-0.2% Er aluminum alloy.

### 3.3. Mechanical Property

Figure 5 shows the Vickers hardness values of the 7055-*x*Er aluminum alloy treated as-cast and with heat treatment. It can be seen that the hardness of the as-cast 7055 aluminum alloy with Er was higher than that of the original as-cast 7055 aluminum alloy. After heat treatment, the hardness value of the 7055 aluminum alloy with a corresponding composition improved except for the addition of 0.6 wt.% Er. After heat treatment, the hardness of adding 0.2 wt.%Er reached the maximum of 168.4 HV, which was 60.4 HV higher than that of the as-cast 7055 aluminum alloy.

Figure 6 shows the tensile curves of 7055 aluminum alloy samples with different Er content before and after heat treatment. Figure 6a,c shows that the elongation of 7055-0.2 wt.%Er in the as-cast state reached the maximum value of 10.54%. The maximum yield strength and tensile strength of 7055-0.2 wt.%Er in as-cast state heat treatment were 542.12 MPa and 577.67 MPa, respectively. Except that the tensile properties of the heat treatment decreased when the Er content was 0.6 wt.%, the tensile properties of the other contents improved after heat treatment (Figure 6b,d).

## 4. Conclusions

In this paper, the changes in the microstructure, phase composition, hardness, and mechanical properties of 7055 aluminum alloy with different Er content before and after heat treatment were studied. The conclusions are as follows:(1)For the 7055-0.2 wt.%Er aluminum alloy after heat treatment, the grain refinement effect was the most obvious; the grain size was 72 μm.(2)Al_8_Cu_4_Er was formed after the addition of Er to 7055 aluminum alloy. The high-density lamellar eutectic *η*-Mg(Zn,Al,Cu)_2_ became thinner. After adding 0.2 wt.%Er, the distribution segregation phenomenon of Mg and Zn was reduced, and the Er element in the 7055-0.2 wt.%Er aluminum alloy after heat treatment was more evenly dispersed than that in the as-cast 7055-0.2 wt.%Er aluminum alloy.(3)At the solution 410 °C × 1 h and aging 150 °C × 12 h, the mechanical properties of the as-cast 7055-0.2 wt.%Er aluminum alloy after heat treatment increased the fastest, and the hardness reached 168.8 HV, which was 60.4 HV higher than that of the 7055 as-cast aluminum alloy. The maximum elongation of 7055-0.2 wt.%Er as the cast was 10.54%, and the yield strength, tensile strength, and elongation of 7055-0.2 wt.%Er as cast heat treatment were 542.12 MPa, 577.67 MPa, and 8.36%, respectively.

## Figures and Tables

**Figure 1 materials-16-05296-f001:**
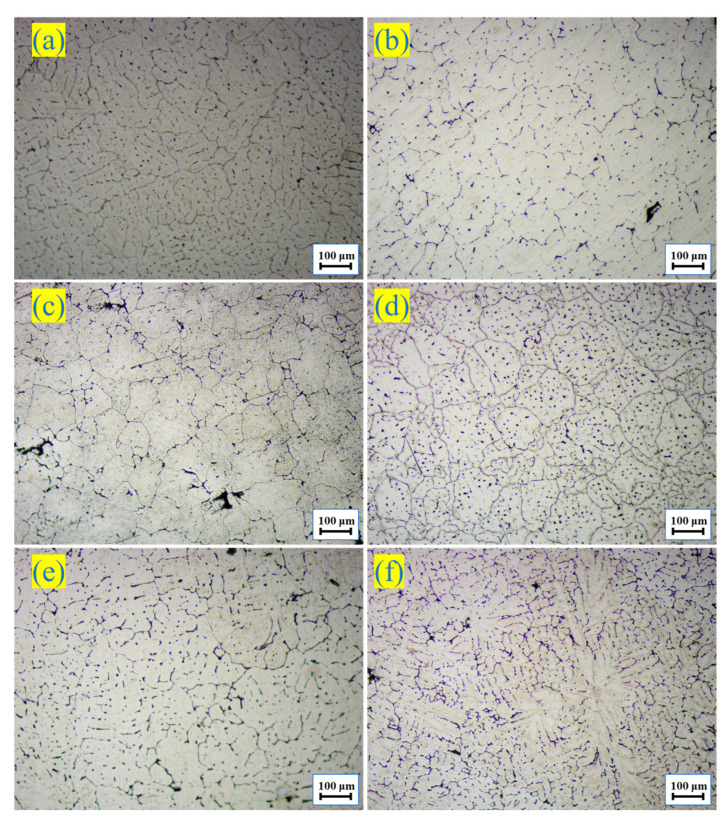
Optical microstructure: (**a**) As-cast 7055-0 wt.%Er; (**b**) As-cast heat-treated 7055-0 wt.%Er; (**c**) As-cast heat-treated 7055-0.2 wt.%Er; (**d**) As-cast heat-treated 7055-0.4 wt.%Er; (**e**) As-cast heat-treated 7055-0.6 wt.%Er; (**f**) As-cast heat-treated 7055-0.8 wt.%Er.

**Figure 2 materials-16-05296-f002:**
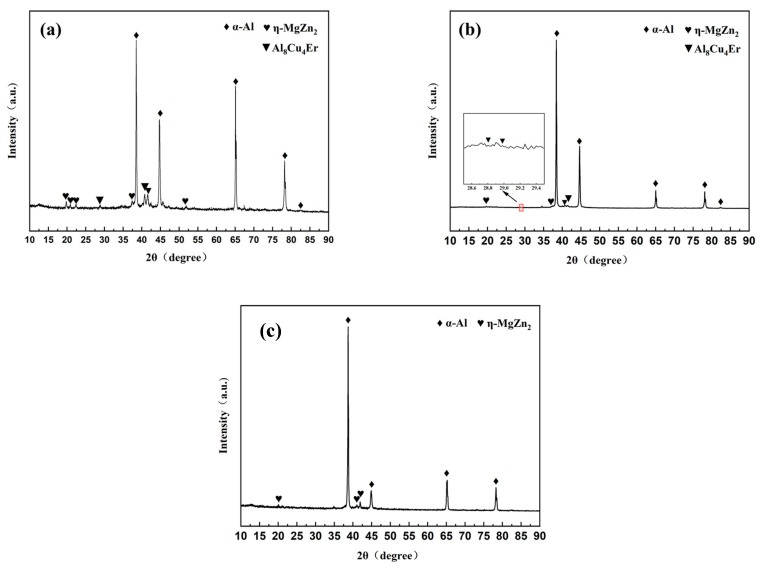
The XRD patterns: (**a**) As-cast 7055-0.2 wt.%Er; (**b**) As-cast heat-treated 7055-0.2 wt.%Er; (**c**) As-cast 7055-0 wt.%Er.

**Figure 3 materials-16-05296-f003:**
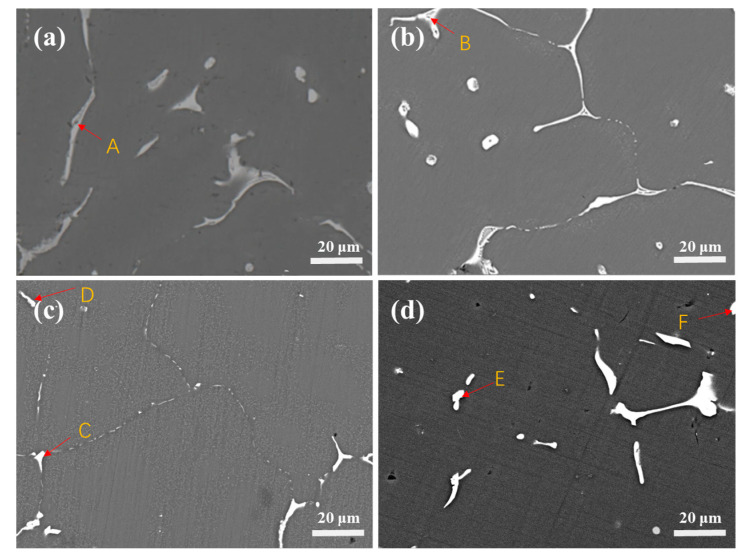
SEM microstructure images: (**a**) As-cast 7055; (**b**) As-cast heat-treated treatment 7055; (**c**) As-cast 7055-0.2 wt.%Er; (**d**) As-cast heat-treated 7055-0.2 wt.%Er; The letters and red arrows in the Figure 3 are the positions of EDS point scanning.

**Figure 4 materials-16-05296-f004:**
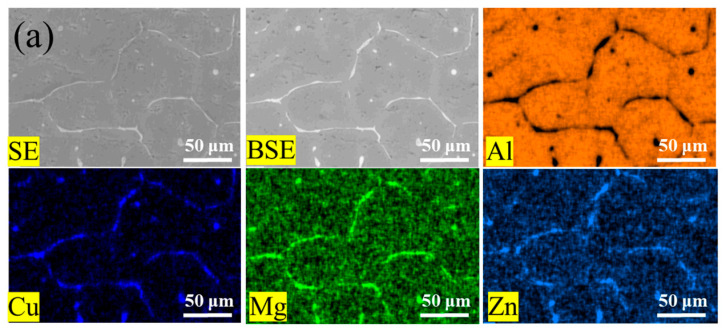
EDS mapping scanning results of alloy samples under different states: (**a**) As-cast 7055; (**b**) As-cast heat-treated 7055; (**c**) As-cast 7055-0.2 wt.%Er; (**d**) As-cast heat-treated 7055-0.2 wt.%Er.

**Figure 5 materials-16-05296-f005:**
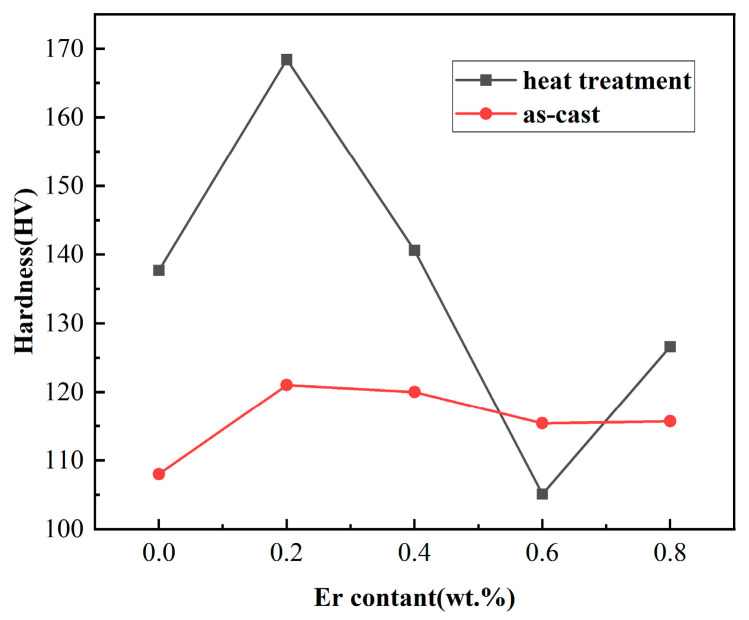
Hardness value curves of the 7055 aluminum alloy with different Er content before and after heat treatment.

**Figure 6 materials-16-05296-f006:**
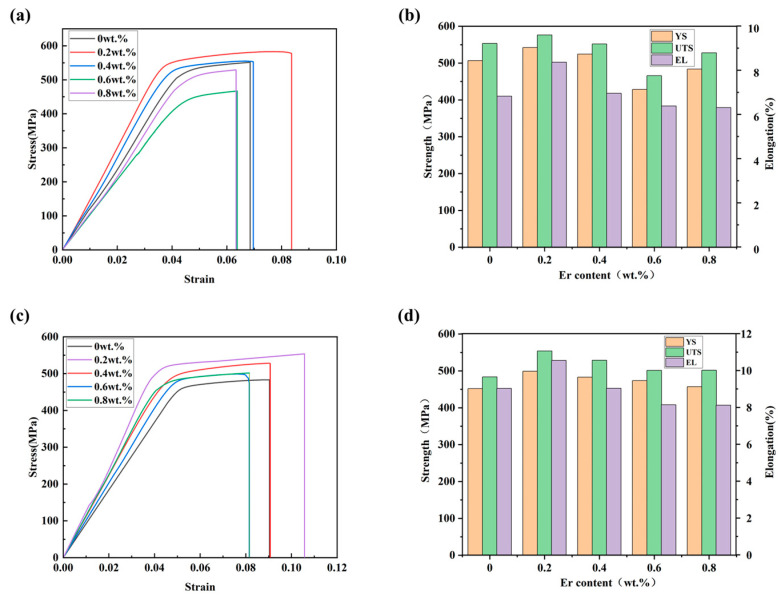
The tensile properties: (**a**) Tensile curve of 7055-*x*Er heat treatment in as-cast state; (**b**) Column chart of yield strength, tensile strength and elongation of 7055-*x*Er for as-cast heat treatment; (**c**) Tensile curve of 7055-*x*Er as-cast; (**d**) Column chart of yield strength, tensile strength and elongation of 7055-*x*Er as-cast.

**Table 1 materials-16-05296-t001:** Chemical composition of the points in Figure 4 determined by EDS analysis (at. %).

Point	Al	Mg	Cu	Zn	Er	Fe	Phase
A	58.98	15.89	21.75	15.21	0	0	*α*-Al+*η*-Mg(Zn,Al,Cu)_2_
B	59.52	18.27	7.95	14.27	0	0	*α*-Al+*η*-Mg(Zn,Al,Cu)_2_
C	58.78	3.39	28.45	6.01	3.36	0	Al_8_Cu_4_Er
D	82.45	1.33	5.25	1.84	0	9.14	*α*-Al+*η*-Mg(Zn,Al,Cu)_2_
E	58.27	1.57	27.92	6.28	4.98	0.98	Al_8_Cu_4_Er
F	25.90	32.61	16.78	24.70	0	0	*α*-Al+*η*-Mg(Zn,Al,Cu)_2_

## Data Availability

Not applicable.

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
