# Peer review of "Study on the Influence of Erbium and Preheating Process on Mechanical Properties of As-Cast 7055 Aluminum Alloy"

_materials, 2023, doi:10.3390/ma16155296_

Round 1

Reviewer 1 Report

The authors investigated the influence of erbium while adding to 7055 Al alloy. The comments are as follows.

1. Authors have already published (10.3390/ma16134846) What is the necessity of this study? How the results differ from the published article? Do you claim adding Sm + Er is poor and only Er is good? No novelty and just an-other experiment.

2. Figure 1 should be removed and provide the ASTM standard for the tensile and hardness study.

3. EBSD analysis shall be done to show grain refinement and boundary (10.1016/j.micron.2022.103401)

4. What is the repeatability and uncertainty of the experiments?

5. Section about porosity study should be included.

6. Future scope and novelty statement should be included.

Extensive English language correction is required.

Author Response

Dear Reviewers:

We are very grateful to Reviewer for reviewing the paper so carefully.

We have carefully considered the suggestion of Reviewer and make some changes.

1.Authors have already published (10.3390/ma16134846) What is the necessity of this study? How the results differ from the published article? Do you claim adding Sm + Er is poor and only Er is good? No novelty and just an-other experiment.

Response: Thanks for the reviewer's kind suggestion. It is necessary to deeply study the single or compound role and mechanism of various rare earths in aluminum alloys. In practical applications, the service environment of different parts is different, and the mechanical properties of materials are different. In addition, the molding methods of different parts are different, and the accommodative heat treatment conditions are different, so the study of adding Er and studying its as-cast heat treatment process has certain application prospects. At present, most articles only point out that a small amount of Er has thinning effect, but there is no detailed study on the heat treatment process of 7055 aluminum alloy containing Er. The research results of this paper show that there is a certain difference in the performance of the optimal state when the content of Er alone and the heat treatment process are compared with the total content of Sm+Er added and the heat treatment process. The optimal solution process for adding Sm+Er was 410℃×2h, and the optimal solution process for adding Er alone was 410℃×1h.

  1. Figure 1 should be removed and provide the ASTM standard for the tensile and hardness study.

Response: Thanks for the reviewer's kind suggestion. Figure 1 has been removed. Hardness standard according to ASTM E92-17, tensile sample standard according to ASTM E646-16.

  1. EBSD analysis shall be done to show grain refinement and boundary (10.1016/j.micron.2022.103401)

Response: Thanks for the reviewer's kind suggestion. EBSD characterization is one of the most important methods to characterize grain size. We also cited K. Baranidharan, S. Thirumalai Kumaran, M. Uthayakumar, P. Parameswaran, D. Arvindha Babu, EBSD analysis of spark plasma sintered SS316-B4C composite, Micron, 166, 2023, 103401. https://doi.org/10.1016/j.micron.2022.103401 article. However, the grain size measured in this paper is the national standard "Metal average grain size determination method" GB/T6397-2017, and the obtained data is credible.

  1. What is the repeatability and uncertainty of the experiments?

Response: Thanks for the reviewer's kind suggestion. The data in this paper are obtained by taking parallel samples. During the stretching 5 parallel samples were taken from each group for the experiment. The uncertainty lies in casting defects,for which all specimens tested for mechanical properties are screened by visual and optical microscopy. Then do density screening. The above has been added to Part 2.2 of the manuscript.

Density screening methods: The weight of the tensile specimen is first measured with an electronic balance. Then the specimen is immersed in a measuring cup with water, the volume of the specimen is calculated. Then their density is calculated. Samples that differ from the theoretical density value by ±0.5% are retained.

  1. Section about porosity study should be included.

Response: Thanks for the reviewer's kind suggestion. The stretched sample has been polish before the stretching test, and the surface is observed under a metallographic microscope without holes before stretching. Then do density screening. And each group took 5 parallel samples for the experiment. The above has been added to Part 2.2 of the manuscript.

  1. Future scope and novelty statement should be included.

Response: Thanks for the reviewer's kind suggestion. It is supplemented in the introduction part of the article: Their biggest consumer is the automotive industry, where they are used in various components such as engine blocks and transmissions in internal combustion engine cars. It can also be used as material for battery car chassis. The novelty is that most of the current papers only point out that a small amount of Er has a refining effect, but there is no detailed study on the heat treatment process of the as-cast 7055 aluminum alloy containing Er. In this paper, a good experimental result is obtained by exploring the appropriate Er content and matching the appropriate heat treatment process.

Comments on the Quality of English Language

Extensive English language correction is required.

Thanks for the reviewer's kind suggestion. We extensive English language correction.

Reviewer 2 Report

see remarks in pdf

Author Response

Dear Reviewers:

We are very grateful to Reviewer for reviewing the paper so carefully.

We have carefully considered the suggestion of Reviewer and make some changes.

-please add also the drawing with dimension markings

Response: Thanks for the reviewer's kind suggestion. Figure 1 has been removed.

-Mg2Zn in (c) ????

Response: Thanks for the reviewer's kind suggestion.This error has been corrected to MgZn2 in this article.

-please instead of SpotX use only X

Response: Thanks for the reviewer's kind suggestion. SpotX has been changed to X.

- you stated that aim of your work is to replace Sc in 7055 alloy, could you discuss the obtained results with simillar results for Scandium from the literature

Response: Thanks for the reviewer's kind suggestion. Perhaps because of the expression problem, in fact, there is no intention in this paper to express the meaning that Er can replace Sc. Because Sc is more commonly used in 7055 aluminum alloy microalloying, the strengthening effect is better. However, due to the high price, we hope to find rare earth elements that are suitable for Er, but can also strengthen aluminum alloys to a certain extent. At present, most articles only point out that a small amount of Er has thinning effect, but there is no detailed study on the heat treatment process of 7055 aluminum alloy containing Er. The purpose of this paper is to find a heat treatment process that can add proper Er content to 7055 aluminum alloy. We have changed the relevant wording in the article to avoid such misunderstandings.

-So for the conclusion, is Er suitable for Sc replacement or not? If yes under which condition???

Response: Thanks for the reviewer's kind suggestion. This paper is not to express that Er can replace Sc, but to point out that Er, as a cheaper rare earth element added to 7055 aluminum alloy, can also improve the mechanical properties of 7055 aluminum alloy after heat treatment process. We have changed the relevant wording in the article to avoid such misunderstandings.

Round 2

Reviewer 1 Report

-

Minor editing of English language required